# Too much to process? Exploring the relationships between communication and information overload and videoconference fatigue

**Benjamin J. Li**[1], **Heng Zhang**[1]*, **Christian Montag**[2]

**1** Wee Kim Wee School of Communication and Information, Nanyang Technological University, Singapore, Singapore, **2** Department of Molecular Psychology, Institute of Psychology and Education, Ulm University, Ulm, Germany

* heng017@e.ntu.edu.sg

**Data Availability Statement:** The data that support the findings of this study are openly available in Figshare at https://doi.org/10.6084/m9.figshare.26772295.v2.

## Abstract

The adoption of videoconferencing has brought significant convenience to people's lives. However, as videoconferencing usage has skyrocketed, it has unveiled a range of side effects, most notably videoconference fatigue (VF). In response, this paper employed the Limited Capacity Model of Motivated Mediated Message Processing (LC4MP) as a theoretical framework to conduct two comprehensive investigations, centering on the impact of verbal communication overload on users' information overload and VF. We conducted two studies to test our propositions and conceptual model. Study One focused on the educational context and comprised a survey with 489 students. In Study Two, we expanded our exploration to the professional use of videoconferencing in two populations: Singapore and Germany. A total of 610 responses were collected in Singapore, with the German sample constituting a total of 948 participants. Results from both studies consistently demonstrated a positive relationship between videoconference frequency and communication overload. Additionally, perceived communication overload was positively associated with information overload and VF. Based on the findings of the two studies, we discuss the theoretical and practical implications and suggest new directions for videoconferencing research.

## Introduction

The pandemic catalyzed unprecedented shifts in human communication patterns, with a notable surge in videoconferencing [1–3]. This technology has become essential for interactions even after the pandemic, especially when in-person meetings are impractical. Therefore, videoconferencing permeates a variety of global settings, from classrooms and workplaces to gatherings between friends and family [4–7]. As remote study and work formats grow, platforms like *Microsoft Teams* and *Zoom* have experienced exponential growth [8]. Videoconferencing for remote study and work is likely to become common practice in the near future [9–11].

**Funding:** This research was funded by a Tier 1 Research Grant from the Singapore Ministry of Education (RG40/22) awarded to BL. The funders had no role in the study design, data collection and analysis, decision to publish, or preparation of the manuscript.

**Competing interests:** The authors have declared that no competing interests exist.

Videoconferencing enables face-to-face communication through the transmission of video and sound, making it a form of computer-mediated communication [12]. According to the media enrichment theory, the degree to which a digital medium emulates face-to-face communication is directly related to its enrichment [13]. From this perspective, videoconferencing stands out as a profoundly enriched communication medium, providing users with abundant nonverbal cues and instantaneous responses to enhance the quality of communication [14]. However, it also has drawbacks. Recent studies identify videoconference fatigue (VF) as one potential concern, where users feel exhausted, disengaged, and anxious after long sessions, and reporting their mental health and productivity severely affected as a result [15–17].

Current research extensively delves into factors that contribute to VF, which can be classified into five primary dimensions: productivity-related, technical, chronemic, psychological, and social [15, p.813]. A few specific contributing factors include mirror anxiety [8, 18], cognitive load [19], restricted physical mobility [19], poor learning experiences [20], and social expectations [21]. Overall, the prevailing focus of these studies highlights the influence of nonverbal communication in videoconferencing on its users. In light of this, Bailenson (2021) introduced the concept of *nonverbal overload*, suggesting that videoconferencing can be mentally taxing due to the need for users to consistently engage with numerous nonverbal cues online, such as maintaining eye contact with multiple attendees or self-monitoring through on-screen self-views [8, 19].

These findings align with key videoconferencing features allowing for the exchange of enriched nonverbal interaction among its users. However, a notable research gap exists concerning verbal communication in videoconferencing. After all, verbal communication is an essential and indispensable component of interpersonal communication [22]. If users experience an overload when processing verbal information in videoconferencing, it is likely that they experience heightened levels of VF.

We seek to investigate this by focusing on communication and information overload caused by verbal communication. Specifically, we draw on the limited capacity model of motivated mediated message processing (LC4MP) to investigate fatigue experienced by users as a result of videoconferencing. Using the LC4MP framework, we investigate how videoconferencing length and frequency impact VF and the mediating roles of perceived information and communication overload. Through two distinct studies, we explore the differences between various groups and the differences under different cultural backgrounds.

## Literature review

### Videoconference fatigue (VF)

As mentioned earlier, prolonged use of videoconferencing can lead to VF. Bennett et al. defined VF as "the degree to which people feel exhausted, tired, or worn out attributed to engaging in a videoconference" [1, p.330]. Other scholars argue that VF may extend beyond mere physical tiredness, encompassing emotional states, motivations, and other factors [23]. Meanwhile, a comprehensive review by Li and Yee [15, p. 813] defines VF as "a non-pathological weariness resulting from videoconferencing, manifesting physically, emotionally, cognitively, and socially".

Demographically, women often report higher levels of VF than men [18, 24, 25]. Additionally, overuse of videoconferencing stands out as a factor that induces fatigue [1, 23]. Previous studies suggest that an escalation in VF can negatively impact individuals' social connections, physical well-being, and mental health [15, p.813, 26]. Specifically, these adverse outcomes include visual distress and irritation of the eyes [23], heightened irritability, tension, and stress [27, 28] and diminished social demands, among others [23]. The negative effects of VF make it

imperative to deepen our understanding of the phenomenon. Nevertheless, as highlighted earlier, while extensive research has examined the causes of VF, they mostly focused on fatigue caused by nonverbal communication in videoconferencing [8, 18, 19]. This study, therefore, seeks to explore the influence of verbal communication on videoconferencing on users' experience of VF.

## Theoretical framework

Extent research on videoconferencing predominantly draws upon theories from two research fields: technology studies and social psychology. These frameworks include the self-presentation theory [29], social presence theory [25], the theory of objective self-awareness [28, 29], media richness theory [21], attention restoration theory [1], and the technology acceptance model [30]. Yet, these theories seem to fall short of fully explaining how VF comes about, particularly as they fail to address how the transmission of information during videoconferencing, particularly the frequency of videoconferencing and amount of information transmitted, can contribute to VF. As videoconferencing demands that participants process an array of information, the limited capacity model of motivated mediated message processing (LC4MP) may offer a fresh perspective in understanding the potential underlying mechanisms of VF.

## The limited capacity model of motivated mediated message processing (LC4MP)

LC4MP utilizes an information-processing approach to analyze the mechanisms of message selection, processing, and subsequent impact [31–33]. The framework assumes that individuals possess limited information processing capacities [32–35]. As people have finite cognitive resources available for perceiving, encoding, understanding, and remembering life events [32], resource scarcity necessitates a cognitive selection process, prioritizing crucial information over less important data [33, 36]. Encoding, storage, and retrieval are the three sub-processes by which LC4MP allocates cognitive resources for message processing [31, 33]. Constraints on cognitive resources can impede the efficacy of these subprocesses, affecting performance metrics and outcomes, including comprehension, enjoyment, and persuasion [36]. Therefore, overburdening a participant during a videoconference meeting, either by excessive communication or information, can drain their cognitive reserves, hampering subsequent engagements or inducing VF.

Furthermore, LC4MP suggests that the activation of the individual motivation system can affect the cognitive processing of media messages [32, 33]. The two motivational systems that underlie human information processing are the appetitive (or approach) system and the aversive (or avoidance) system [32]. The appetitive system primarily aims for information acquisition, emphasizing the encoding of details about incoming stimuli and their surrounding environment [32, 33, 37]. According to LC4MP, activating the appetitive system enhances the automatic allocation of resources for encoding and storage; meanwhile, as stimuli become more enticing, both the resource allocation and levels of appetitive activation intensify [33, 38]. By contrast, the primary goal of the aversive system is protection rather than information acquisition [31]. At low levels of aversive system activation, it is curtailed to identify potential negative stimuli, directing automatic resource allocation towards encoding [31]. As the level of aversive system activation increases, cognitive focus shifts to potential responses to challenges, such as defense or withdrawal, prioritizing resource allocation to retrieval [33, 38].

According to Lang (2017), LC4MP can applied across multiple mediums, user types, contexts, and message types [33]. LC4MP has been employed in various disciplines, including health communication, political communication, persuasion and social influence, gender roles

and stereotypes, ethics, journalism, media multitasking, educational media, and entertainment [36]. Given that videoconferencing fundamentally requires individuals to undergo information processing during virtual communication, LC4MP appears to be an appropriate framework to undergird this study.

In the context of the present study, if users perceive an excessive amount of information and communication demands, they risk facing communication overload and information overload. The perception of this threat is contingent on the participants' discernment within the videoconferencing environment: when overloaded, they may lack the necessary cognitive resources to process the information adequately. The ensuing sections will delve deeper into how both communication and information overload contribute to VF, as well as their potential antecedents.

## Communication overload

The concept of communication overload pertains to the excessive burden experienced by individuals due to sustained and intensive interactions in the communication domain [39]. It is defined as the excessive communication demands that surpass an individual's ability to cope [40]. This study primarily centers on verbal communication, encompassing the verbal expressions and interactions of users during videoconferencing. Communication overload can disrupt users' routines, impede their focus and prompt them to cease current tasks [41–43]. Stephens et al. (2017) identify extensive use of information and communication technologies (ICTs) as one of the primary contributing factors to communication overload [44].

Videoconferencing, which simulates real-time face-to-face interactions, requires users to continuously engage with one another. Every exchange within this medium necessitates participants to assimilate information and reciprocate with feedback. In this context, individuals tend to respond quickly to each other's input, aiming to adhere to societal norms [45]. Hence, an increase in both the length and frequency of videoconferencing may result in a corresponding rise in users' communication requirements. The following hypotheses are proposed:

**H1**: Videoconference frequency is positively associated with communication overload.

**H2**: Videoconference length is positively associated with communication overload.

According to Pang and Ruan (2023), individuals may suffer feelings of tension and mental fatigue when confronted with communication overload when using mobile media [46]. Based on LC4MP, the human capacity for information processing is limited [32, 33]. Without timely replenishment of cognitive resources, individuals might not have sufficient capacity to handle their communication needs, especially when these resources are already drained. When confronted with challenging social communication without adequate resources or the right communication skills, individuals can become overwhelmed, leading to fatigue and anxiety [47]. Thus, we propose the following hypothesis:

**H3**: Communication overload from videoconferencing is positively associated with videoconference fatigue.

## Information overload

In computer-mediated environments, information overload refers to the overwhelming state individuals experience when exposed to an excessive amount of information across different media platforms, challenging their capacity to assimilate and manage it [48, 49]. Both information quantity and quality contribute to this overload, especially when there is a large volume and poor quality of information [50]. When individuals experience information overload, they

are often prone to experience confusion, psychological stress, and other emotional states [46]. Prior research on information overload in videoconferencing has focused on nonverbal communication cues, such as gestures and eye contact [15, p.813]. However, the focus of information overload in this study is on the verbal information generated when users communicate with other participants in videoconferencing, which predominantly pertains to verbal communication. When individuals frequently engage in prolonged videoconferencing sessions, they will receive a greater volume of verbal information. Simultaneously, their heightened communication needs correlate with the increased volume of information they have to sift through. Therefore, we propose the following hypotheses:

**H4**: Videoconference frequency is positively associated with information overload.

**H5**: Videoconference length is positively associated with information overload.

**H6**: Communication overload from videoconferencing is positively associated with information overload.

Based on LC4MP, individuals typically have restricted information processing capacities [32, 33]. When faced with a significant volume of content, their decision-making quality can be compromised [32, 46]. This can subsequently result in decreased work efficiency, consequently impacting user fatigue during videoconferencing [15]. Moreover, encountering irrelevant or misaligned information can intensify the sensation of overload [46]. During videoconferencing, users may be drawn into off-topic discussions, taxing their cognitive resources and leading to exhaustion. Such circumstances might activate a user's aversion system, shifting focus from information gathering to self-preservation [32]. As a result, users may prioritize coping with overactive engagement, further reducing work efficiency, eliciting negative emotions, and exacerbating VF. We hence propose the following hypothesis:

**H7**: Information overload from videoconferencing is positively associated with videoconference fatigue.

The proposed conceptual model is presented in Fig 1.

## Study One: Student population

Although online classes existed before the COVID-19 pandemic, they were not the dominant form of learning. The pandemic mandated students to transition from traditional face-to-face education to videoconferencing for schoolwork to ensure uninterrupted educational progress [51]. In the context of VF, it was observed that individuals who use videoconferencing for educational purposes experience higher levels of fatigue than those who primarily use it for work-

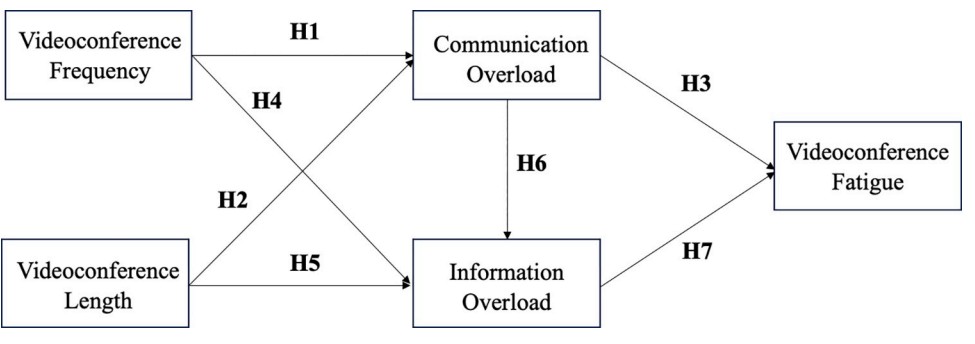

**Fig 1. Proposed conceptual model.**

related activities [2]. In addition, VF can significantly impact students' grades and mental well-being [52]. Considering the important role VF plays in the educational context, our first study focused on students as our population of interest.

Moreover, among the student population, we focus specifically on secondary school students. The reasons are as follows: firstly, according to reports from the Singaporean education authorities, approximately 96% of secondary school students participated in full home-based learning during the COVID-19 pandemic [53]. Compared to university students, secondary school students had almost no prior experience with online courses before the pandemic. Many university students might have had online education experience before COVID-19, so attending classes via videoconferencing is not unfamiliar to them, but it is a completely new experience for secondary school students. Secondly, secondary school students have more intensive course schedules compared to university students. They need to use videoconferencing regularly every day for their classes, making them more prone to VF. To address technical challenges, the Singaporean government provided each secondary school student with a tablet, whereas university students typically already have their own electronic devices [54]. In summary, using videoconferencing as a teaching method is more challenging for secondary school students.

## Method

**Procedure and participants.**   An online nationwide survey via *Qualtrics* was conducted from 3 October to 30 November 2022 with 489 students from four secondary schools in Singapore. *Qualtrics* is a highly regarded and extensively utilized survey platform that supports the comprehensive creation, distribution, and analysis of surveys [55]. It offers an online format, making it easily accessible to participants using various digital devices, including computers, tablets, and smartphones. Administrators from the schools facilitated the data collection.

The study was approved by the Institutional Review Board of Nanyang Technological University. Participation was entirely voluntary. All respondents provided their informed consent prior to data collection, and consent was also obtained from the parents of respondents who were under the age of 21. Before the survey began, all participants were given sufficient time to read the Information Sheet and sign it. During the experiment, all participants were made aware about their rights to withdraw at any time or choose not to answer any questions that made them feel uncomfortable.

**Measures.**   *Videoconference frequency*. Participants were asked to indicate the number of videoconferences they participate in on a typical day ($M = 1.77$, $SD = 0.94$).

*Videoconference length*. Participants were asked to indicate the length of a typical videoconference session. This was measured using a multiple-choice question with options ranging from 1 = less than 15 minutes to 5 = one hour and above ($M = 4.47$, $SD = 0.85$).

*Communication overload*. We adopted three items from Lee et al. and rephrased them to fit the context of videoconferencing [56] ($\alpha = .82$, $M = 2.65$, $SD = 0.87$). The questions were measured on a 5-point Likert scale item (1 = not at all, 5 = extremely).

*Information overload*. Three items were similarly adopted from Lee et al., with the context changed to videoconferencing [56] ($\alpha = .87$, $M = 2.94$, $SD = 0.93$). Questions were measured on a five-point Likert-type scale (1 = not at all, 5 = extremely).

*Videoconference fatigue*. Videoconference fatigue was measured using the Zoom Exhaustion & Fatigue (ZEF) scale [23]. The fifteen questions were rated on a five-point Likert scale (1: Not at all; 5: Extremely). The items contain five dimensions: general, visual, social, motivational, and emotional ($\alpha = .94$, $M = 2.78$, $SD = 0.83$). Fauville et al. presents a discussion on the conceptual differences between the factors and a more detailed explanation behind the individual items [23].

**Table 1. Measurement items, descriptive statistics and analysis results for Study One.**

| | Items | M(SD) | $\lambda$ | AVE | CR |
|---|---|---|---|---|---|
| **Frequency** | On a typical day, how many videoconferences do you participate in? | 1.77 (.94) | | | |
| **Length** | Based on your personal experience, how long does a typical videoconference last? | 4.47 (.85) | | | |
| **Communication Overload** | | | | .73 | .89 |
| Commload1 | I receive too many videoconferencing requests. | 2.35 (.90) | .85 | | |
| Commload2 | I feel like I must attend more videoconferences than I would like to. | 2.80 (1.10) | .88 | | |
| Commload3 | I often feel overloaded with communication from videoconferencing. | 2.79 (1.05) | .84 | | |
| **Information Overload** | | | | .80 | .92 |
| Infoload1 | I am often distracted by the excessive amount of information in videoconferencing. | 2.92 (1.01) | .88 | | |
| Infoload2 | I find that I am overwhelmed by the amount of information that I process from videoconferencing. | 2.86 (1.03) | .93 | | |
| Infoload3 | I find it difficult to process all the information during videoconferencing. | 3.03 (1.08) | .87 | | |
| **Videoconference Fatigue** | | | | .54 | .94 |
| Genfat1 | How tired do you feel after videoconferencing? | 3.25 (.99) | .75 | | |
| Genfat2 | How exhausted do you feel after videoconferencing? | 3.15 (1.00) | .78 | | |
| Genfat3 | How mentally drained do you feel after videoconferencing? | 3.27 (1.05) | .77 | | |
| Visfat1 | How blurred does your vision get after videoconferencing? | 2.12 (1.03) | .63 | | |
| Visfat2 | How irritated do your eyes feel after videoconferencing? | 2.43 (1.09) | .63 | | |
| Visfat3 | How much do your eyes hurt after videoconferencing? | 2.34 (1.10) | .66 | | |
| Socfat1 | How much do you tend to avoid social situations after videoconferencing? | 2.39 (1.24) | .70 | | |
| Socfat2 | How much do you want to be alone after videoconferencing? | 2.78 (1.33) | .76 | | |
| Socfat3 | How much do you need time alone by yourself after videoconferencing? | 2.75 (1.27) | .75 | | |
| Motfat1 | How much do you dread having to do things after videoconferencing? | 2.95 (1.18) | .73 | | |
| Motfat2 | How often do you feel like doing nothing after videoconferencing? | 3.46 (1.17) | .64 | | |
| Motfat3 | How often do you feel too tired to do other things after videoconferencing? | 3.24 (1.15) | .77 | | |
| Emofat1 | How emotionally drained do you feel after videoconferencing? | 2.65 (1.17) | .83 | | |
| Emofat2 | How irritable do you feel after videoconferencing? | 2.46 (1.12) | .77 | | |
| Emofat3 | How moody do you feel after videoconferencing? | 2.41 (1.15) | .76 | | |

*Note.* AVE = average variances extracted; CR = composite reliability.

Table 1 shows the items used and provides a summary of descriptive statistics from the analysis.

## Data analysis

Structural equation modeling was utilized to validate the proposed model. In accordance with the methodology proposed by Hu and Bentler [57], the adequacy of fit for both the measurement model and the structural model was evaluated by using the following information criteria: comparative fit index (CFI) > .95, 90% confidence interval of the root mean square error of approximation (RMSEA) < .05, and standardized root mean square residual (SRMR) < .08. The measurement model showed a good fit, $\chi2$ [73] = 212.89, p < .001; CFI = .95; RMSEA = .06, SRMR = .05, with the values considered acceptable [58]. The measurement model and residual covariances were not altered. The measurement paradigm was employed to evaluate the construct validity and reliability (see Table 1). In terms of reliability, every composite reliability (CR) value surpassed .70. With regard to validity, all average variances extracted (AVEs) exceeded .50.

## Results

Respondents comprised 185 (37.8%) males and 304 (62.2%) females, with a mean age of 17.00 years (*SD* = 4.85). As Singapore is a multiracial country, our sample consisted of respondents

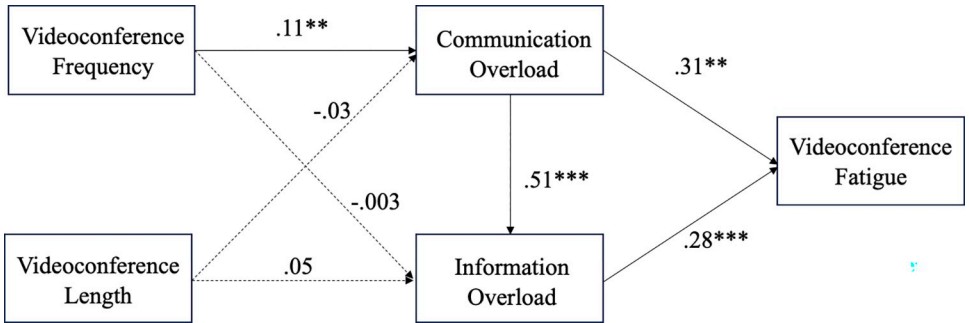

**Fig 2. Structural equation model for Study One.**

from various ethnic groups, including Chinese (85.3%), Malay (3%), Indian (8.5%), Eurasian (0.2%) and other ethnicities (3%).

Fig 2 and Table 2 displays the coefficients of each path. Videoconference frequency was found to be positively associated with communication overload, thus supporting H1 ($\beta$ = .09, $p$ = .01) However, there was no significant relationship between videoconference length and communication overload. Thus, H2 ($\beta$ = -.02 $p$ = .56) was not supported. In support of H3 ($\beta$ = .44, $p < .001$), communication overload was positively associated with videoconference fatigue. No significant relationship was found between videoconference frequency and information overload, nor between videoconference length and information overload. Therefore, H4($\beta$ = .01, $p$ = .56) and H5($\beta$ = .04, $p$ = .30) were not supported. In support of H6($\beta$ = .69, $p < .001$) communication overload was positively associated with information overload. Information overload was positively associated with videoconference fatigue, thereby supporting H7($\beta$ = .28, $p < .001$).

## Discussion

In Study One, we observed a positive relationship between videoconference frequency and communication overload. However, videoconference frequency showed no significant connection to information overload, while videoconference length did not influence either form of overload. A possible reason is that during videoconference lessons, students are often

**Table 2. Results of hypotheses testing for Study One.**

|  | $\beta$ | $p$ | 95% CI |
|---|---|---|---|
| VCFreq → Commload | .09 | .01 | [.022, .160] |
| VCLength → Commload | -.02 | .56 | [-.098, .053] |
| VCFreq → Infoload | .01 | .74 | [-.059, .084] |
| VCLength → Infoload | .04 | .30 | [-.036, .119] |
| Commload → Infoload | .69 | < .001 | [.555, 0.823] |
| Commload → VF | .44 | < .001 | [.310, .564] |
| Infoload → VF | .28 | < .001 | [.184, .375] |
| Age → VF | -.01 | .03 | [-.239, -.013] |
| Gender → VF | .01 | .03 | [.001, .024] |

*Note*. VCFreq = VC frequency; Commload = communication overload; Infoload = information overload; 95% CI = bias corrected bootstrapping 95% confidence interval (bootstrapping subsample = 5000). Gender and age were controlled for all endogenous variables. For clarity, these covariates were only displayed age and gender for VCfatigue.

passive recipients of information, even during long sessions with minimal communication requirements [59]. In other words, during online classes conducted via videoconferencing, students may find themselves listening to the teacher more frequently than actively participating, especially when there is limited time allocated for questions and answers. Given that not every class necessitates each student to respond to questions, there is typically minimal communication required among students in online classes. Moreover, prior studies have demonstrated that students are highly susceptible to mind wandering during videoconferencing [52, 59]. Hence, fewer cognitive resources are spent on videoconferencing, translating to less information overload. We further found that both communication overload and information overload were significantly related to VF, implying that an increase in communication and information overload can contribute to VF. These findings align with the LC4MP model, suggesting that individuals experience fatigue when their cognitive resources are over-capacity. To further validate our model, Study Two investigates the proposed relationships among the working demographic.

## Study Two: Working population (Singapore and Germany)

Results of Study One showed that among students, communication overload is positively related to information overload and VF. While students have been shown to suffer from VF, the significance of VF in the workplace—especially in the post-pandemic landscape—should not be overlooked. Recent data suggests that over 87% of employees would choose remote work if given the opportunity [9], and more than 66% of senior employees would prefer to resign rather than accept full-time office employment [60]. Therefore, Study Two focuses on working professionals, examining the plausibility of the model beyond videoconferencing in education.

Furthermore, significant disparities in the adoption and diffusion of technology have been observed across countries [61, 62]. One plausible explanation is that the cultural norms of different countries shaped different attitudes towards technology use, while another explanation is that social and governmental systems may influence technology adoption [63]. Notably, multiple studies have underscored the potential limitation of transposing theories developed in one culture to another [64, 65]. With these considerations, for Study Two, we selected Singapore and Germany as our sampling population. It is conceivable that differences in these two countries may lead to varying outcomes regarding the previously proposed hypotheses (H1-H7). Hence, in addition to the hypotheses, we propose the following research question:

RQ1: How does the impact of communication overload and information overload on VF differ between the working populations of Singapore and Germany?

### Method

**Procedure and participants.**   Online surveys were conducted in both countries using the *Qualtrics* survey platform from 6 March to 30 May 2023. All participants were over 21 years old and had prior experience using videoconferencing. Participants who did not meet these criteria were not included in the study population for both countries.

An English questionnaire was utilized in Singapore to assess the measures, with the questions translated into German for the second sample. In order to verify the semantic consistency of the English and German questions, the translations underwent a two-way check by the authors. The methodology and survey had received approval from the Institutional Review Board of Nanyang Technological University before the commencement of data collection. In adherence to ethical standards, participants were fully informed of the study's procedures,

offered the option of choosing to withdraw from the research, had their data anonymized, and were requested to provide informed consent before the commencement of the survey.

## Measures

Participants completed the questionnaire with the same measures of videoconference frequency, videoconference length, communication overload, information overload and videoconference fatigue (ZEF) and demographic questions as in Study One. Table 3 shows the items, descriptive and analysis statistics.

## Data analysis

We used multigroup structural equation modeling (SEM) to explore H1 to H7 in the two samples and the potential differences between the countries (RQ1). We first performed a confirmatory factor analysis (CFA) to test our multi-item measures for communication overload,

**Table 3. Measurement items, descriptive statistics and analysis results for Study Two.**

|  |  | Singapore | | Germany | |
|---|---|---|---|---|---|
|  | **Items** | **M (SD)** | **λ** | **M (SD)** | **λ** |
| **Frequency** | On a typical day, how many videoconferences do you participate in? | 2.38 (1.56) |  | 2.00 (1.25) |  |
| **Length** | Based on your personal experience, how long does a typical videoconference last? | 2.97 (1.16) |  | 2.50 (1.30) |  |
| **Communication Overload** |  | 3.12 (0.95) |  | 2.24 (1.02) |  |
| (SG: AVE = .75, CR = .90; GE: AVE = .73, CR = .89) |  |  |  |  |  |
| Commload1 | I receive too many videoconferencing requests. | 2.96 (1.12) | .87 | 2.1 (1.14) | .85 |
| Commload2 | I feel like I must attend more videoconferences than I would like to. | 3.14 (1.12) | .87 | 2.29 (1.24) | .91 |
| Commload3 | I often feel overloaded with communication from videoconferencing. | 3.27 (1.06) | .86 | 2.33 (1.21) | .79 |
| Information Overload |  | 3.34 (0.89) |  | 2.62 (1.08) |  |
| (SG: AVE = .75, CR = .90; GE: AVE = .80, CR = .92) |  |  |  |  |  |
| Infoload1 | I am often distracted by the excessive amount of information in videoconferencing. | 3.31 (1.06) | .88 | 2.61 (1.22) | .89 |
| Infoload2 | I find that I am overwhelmed by the amount of information that I process from videoconferencing. | 3.32 (1.03) | .88 | 2.50 (1.21) | .91 |
| Infoload3 | I find it difficult to process all the information during videoconferencing. | 3.38 (.99) | .83 | 2.76 (1.20) | .88 |
| Videoconference Fatigue |  | 2.72 (0.99) |  | 2.12 (0.92) |  |
| (SG: AVE = .68, CR = .97; GE: AVE = .66, CR = .98) |  |  |  |  |  |
| Genfat1 | How tired do you feel after videoconferencing? | 3.00 (1.09) | .80 | 2.11 (1.12) | .87 |
| Genfat2 | How exhausted do you feel after videoconferencing? | 2.92 (1.10) | .81 | 2.19 (1.12) | .87 |
| Genfat3 | How mentally drained do you feel after videoconferencing? | 2.96 (1.16) | .83 | 2.18 (1.15) | .86 |
| Visfat1 | How blurred does your vision get after videoconferencing? | 2.47 (1.22) | .78 | 2.11 (1.13) | .77 |
| Visfat2 | How irritated do your eyes feel after videoconferencing? | 2.53 (1.21) | .83 | 2.24 (1.12) | .74 |
| Visfat3 | How much do your eyes hurt after videoconferencing? | 2.45 (1.22) | .82 | 2.02 (1.11) | .74 |
| Socfat1 | How much do you tend to avoid social situations after videoconferencing? | 2.67 (1.22) | .81 | 1.96 (1.15) | .82 |
| Socfat2 | How much do you want to be alone after videoconferencing? | 2.83 (1.26) | .77 | 2.21 (1.24) | .79 |
| Socfat3 | How much do you need time alone by yourself after videoconferencing? | 2.84 (1.25) | .82 | 2.21 (1.18) | .80 |
| Motfat1 | How much do you dread having to do things after videoconferencing? | 2.86 (1.25) | .83 | 2.05 (1.19) | .81 |
| Motfat2 | How often do you feel like doing nothing after videoconferencing? | 2.99 (1.22) | .77 | 2.29 (1.12) | .81 |
| Motfat3 | How often do you feel too tired to do other things after videoconferencing? | 2.83 (1.23) | .86 | 2.19 (1.10) | .82 |
| Emofat1 | How emotionally drained do you feel after videoconferencing? | 2.68 (1.23) | .88 | 2.01 (1.11) | .81 |
| Emofat2 | How irritable do you feel after videoconferencing? | 2.43 (1.23) | .87 | 1.96 (1.12) | .82 |
| Emofat3 | How moody do you feel after videoconferencing? | 2.39 (1.25) | .85 | 2.1 (1.13) | .84 |

*Note*. SG = Singapore; GE = Germany; AVE = average variances extracted; CR = composite reliability.

**Table 4. Measurement and structural equation models for Study Two.**

| Structural Model | X2 | df | p | CFI | RMSEA | SRMR |
|---|---|---|---|---|---|---|
| SG$_{SEM}$ | 233.13 | 75 | < .001 | .97 | .059 | .03 |
| GES$_{EM}$ | 622.79 | 75 | < .001 | .93 | .08 | .057 |
| MULTI$_{SEM}$ | 927.49 | 158 | < .001 | .939 | .079 | .053 |
| MULTI-CONST$_{SEM}$ | 997.28 | 166 | < .001 | .934 | .08 | .057 |

information overload, and VF. A set of fit indices [57] revealed a good fit for the CFA in both samples (model SG$_{CFA}$ and GE$_{CFA}$, Table 4). The measures in both samples also showed good internal consistency (see Table 3). In terms of reliability, each composite reliability (CR) value surpassed 0.70. Regarding validity, all average variances extracted (AVEs) exceeded 0.50.

Further, the multigroup model showed a good fit to the data (MULTI$_{SEM}$, Table 4). We found weak invariance, as the difference between the two models in the comparative fit index (CFI) was less than 0.01 and in the root mean square error of approximation (RMSEA) was less than .015 [66]. We tested for differences in paths between the Singaporean and German samples by comparing this (unconstrained) multigroup model (MULTI$_{SEM}$) with a constrained model in which the regression coefficients were held equal across the two countries (MULTI-CONST$_{SEM}$). The difference in goodness of fit between the models was significant ($\chi2 = 541.40$, $p < .001$), implying significant differences in some paths between the Singapore and Germany models. Thus, to answer *RQ1*, we identified the paths causing the difference by constraining the regression paths (one at a time) and testing for differences in goodness of fit.

## Results

A total of 610 responses were collected in Singapore, with a mean age of 43.67 years ($SD = 12.24$). This comprised 284 males (46.6%) and 326 females (53.4%). The German sample constituted a total of 948 participants with a mean age of 43.81 years ($SD = 12.45$), with 475 males (50.1%) and 473 females (49.9%). Table 5 presents a breakdown of the study sample.

In samples from both countries, we found that videoconferencing frequency was positively associated with communication overload, thus supporting H1. However, the relationship is significantly stronger in the German ($ß = .20$, $p < .001$) sample than in Singapore ($ß = .13$, $p < .001$; $\Delta ß = -.07$, $p = .04$). There was no significant relationship between videoconference length and communication overload for both countries, hence H2 was not supported ($ß_{Singapore} = .05$, $p = .07$; $ß_{Germany} = -.01$, $p = .62$; $\Delta ß = .07$, $p = .08$). In support of H3 ($ß_{Singapore} = .56$, $p < .001$; $ß_{Germany} = .31$, $p < .001$; $\Delta ß = .25$, $p = .02$), communication overload was positively associated with VF in both samples, with the relationship being significantly stronger in the Singapore sample than in the German sample. Interestingly, contrary to H4 ($ß_{Singapore} = -.07$, $p < .001$; $ß_{Germany} = -.02$, $p < .001$; $\Delta ß = .12$, $p < .001$), there was a significant inverse relationship between videoconference frequency and information overload for both countries. There was no significant relationship between videoconference length and information overload in the Singapore sample, but an inverse relationship exists in the German sample. Hence, H5 ($ß_{Singapore} = .01$, $p = .55$; $ß_{Germany} = -.08$, $p < .001$; $\Delta ß = .10$, $p = .003$) was not supported. In

**Table 5. Demographic data of Singapore and German samples.**

| Country | Total Responses | Mean Age (Years) | Standard Deviation | Male | Female |
|---|---|---|---|---|---|
| Singapore | 610 | 43.67 | 12.24 | 284 (46.4%) | 326 (53.4%) |
| Germany | 948 | 43.81 | 12.45 | 475 (50.1%) | 473 (49.9%) |

**Table 6. Standardized structural model paths for Study Two.**

| | Singapore | | Germany | | Path difference | |
| --- | --- | --- | --- | --- | --- | --- |
| Path | *B* | *SE* | *B* | *SE* | *ΔB* | *SE* |
| H1: VCFreq → Commload | .13*** | .002 | .20*** | .03 | -.07* | .03 |
| H2: VCLeng → Commload | .05 | .03 | -.01 | .02 | .07 | .04 |
| H3: Commload → VF | .56*** | .09 | .31*** | .05 | 0.25* | .10 |
| H4: VCFreq → Infoload | -.07*** | .02 | -.02*** | .03 | 0.12*** | .03 |
| H5: VCLeng → Infoload | .01 | .02 | -.08*** | .02 | 0.10** | .03 |
| H6: Commload → Infoload | .85*** | .05 | .91*** | .05 | -0.06 | .06 |
| H7: Infoload → VF | .23** | .09 | .36*** | .04 | -0.14 | .10 |

*Note. 1.* VCFreq = VC frequency; Commload = communication overload; Infoload = information overload; VF = videoconferencing fatigue; SG = Singapore; GE = Germany.

\* < .05

\*\* < .01

\*\*\* < .001

support of H6 ($\beta_{\text{Singapore}}$ = .85, $p$ < .001; $\beta_{\text{Germany}}$ = .91, $p$ < .001; $\Delta\beta$ = -.06, $p$ = .27), communication overload was positively associated with information overload for both countries. Results also showed a positive relationship between information overload and VF for both countries. Therefore, H7 ($\beta_{\text{Singapore}}$ = .23, $p$ < .01; $\beta_{\text{Germany}}$ = .36, $p$ < .001; $\Delta\beta$ = -.14, $p$ = .17) was supported. In answering RQ1, we found significant differences among the paths between Germany and Singapore, with most tested relationships significantly stronger in the Singapore sample. Hence, the impact of videoconference use on VF is not the same in both countries. Table 6 displays the coefficients of each path for Singapore and Germany and the difference in path between the two countries, while Figs 3 and 4 show the structural equation models for Singapore and Germany.

## Discussion

In Study Two, our focus shifted to working professionals who utilize videoconferencing for work-related purposes. Our analyses revalidated the robustness of our proposed model and showed differences in the relative strengths and directions of the relationships between the antecedents and experience of VF in Singapore and Germany. Next, we will delve into the results, especially emphasizing the main distinctions between the two samples.

Firstly, results from both Singapore and Germany demonstrate that videoconference frequency directly influences communication overload. Interestingly, this correlation is notably

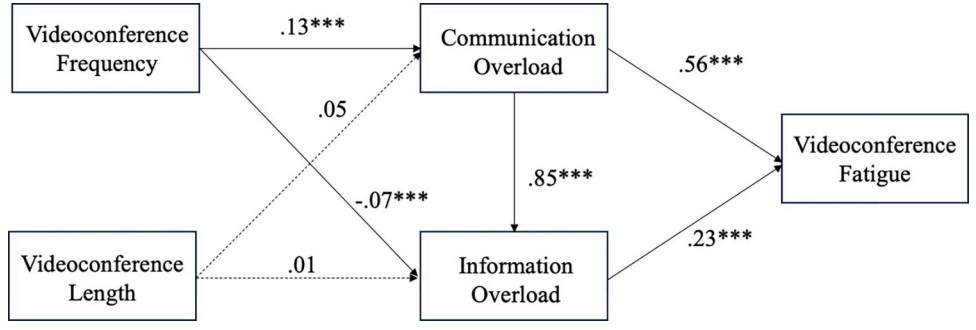

**Fig 3. Structural equation model for Singapore sample in Study Two.**

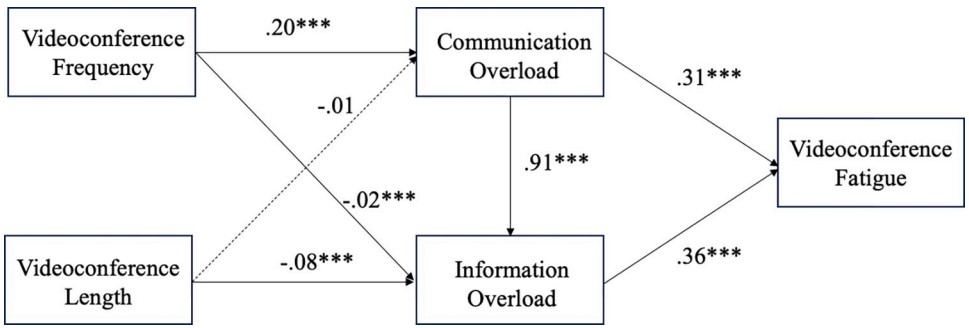

**Fig 4. Structural equation model for Germany sample in Study Two.**

stronger in Germany than in Singapore, likely due to the contrasting cultural foundations in these two countries. Eastern and Western cultures typically differ in their emphasis on collectivist or individualist values [67]. In our study, Germany leans towards individualism, while Singapore prioritizes collective values [68]. Individuals in individualistic societies may be more inclined to express their thoughts during videoconferencing [69, 70], resulting in increased communication needs. By contrast, those in collectivist societies may be more willing to conform to group arrangements, minimizing individual expressions. Surprisingly, the length of videoconferencing does not appear to influence communication overload. This might be attributed to the nature of extended sessions where participants focus more on listening and task adherence than voicing personal opinions.

Additionally, we observed a negative correlation between videoconference frequency and information overload in both samples, contrasting with earlier studies that indicated a positive relationship [8, 71]. One plausible explanation for this discrepancy is that videoconferencing has gradually integrated into people's work routines: the data collection for this study occurred after the pandemic when videoconferencing had already become a habitual practice [72, 73]. Moreover, from a practical standpoint, it is possible that higher frequency of videoconferencing among colleagues can lead to more effective working relationships, as the greater interaction between communication partners can help them develop a stronger understanding of one another's perspectives and working styles. Frequent videoconference meetings also allow colleagues more opportunities to clarify doubts and reduce potential misunderstandings. According to LC4MP, individuals typically go through three steps when processing information: encoding, storage, and retrieval [32, 33]. Indeed, when people have more opportunity to communicate with their colleagues, they can inquire about potential uncertainties during the encoding phase, avoiding the need to expend additional cognitive resources.

Data from the Singapore sample showed no significant relationship between videoconference length and information overload, while an inverse relationship was shown in the German sample. The distinct cultures of the two countries may contribute to this contrast. German participants' tendency for expressive communication may potentially mitigate cognitive overload [74]. In Western culture, where individuals often hold independent ideas, a task with limited details may result in varied interpretations among workers. The longer the meeting lasts, the more opportunity individuals have to express their understanding of a task and avoid disagreements. Although this process provides participants with more information, it helps clarify task requirements, reducing potential misunderstandings and avoiding additional cognitive load caused by unclear information. On the other hand, Singaporeans, being influenced by Asian values, are more inclined to adhere to organizational arrangements [75]. In Asian work cultures with a more rigid hierarchical structure, leaders often directly assign tasks, and

employees readily accept these directives, contributing to error avoidance in the workplace [75]. Therefore, this could lead to employees developing a mindset of compliance with the leader's arrangements, meaning that regardless of the videoconferencing's duration, the purpose of the meeting is to adhere to the leader's arrangements, rather than to engage in discussion [76].

Data from both Singapore and Germany reveal a positive relationship between communication overload and information overload. Notably, this correlation is stronger in Singapore than in Germany. According to LC4MP, communication involves continuous exchange of information via media [32, 33, 36]. In the context of videoconferencing, when individuals experience heightened communication needs during videoconferencing, they not only receive a substantial volume of information but also need to process the information and respond rapidly, thereby inducing information overload.

In conclusion, data from Study Two show positive relationships between communication overload, information overload, and VF. However, there are some differences between the two countries. In Singapore, communication overload showed a greater impact on VF. This may be due to its dominant collectivist culture, where people tend to be more compliant rather than seeking to express their own opinions [77]. If they are continually asked to express themselves during a videoconferencing, they may inherently feel a certain level of pressure. Conversely, in the individualistic culture of the German sample, information overload seems to have a more pronounced effect on VF. While self-expression in conferences might be commonplace, managing a deluge of information can lead to heightened fatigue.

## General discussion

In summary, this paper attempts to validate a proposed theoretical model through two distinct investigations, one in the educational context and one in the context of the workplace. It also explores the model with samples from two countries: Singapore and Germany. The application of LC4MP introduces a novel perspective to understand the psychological mechanism of VF. Notably, in contrast to many prior studies that focus on nonverbal communication, our work centers on verbal communication, elevating its theoretical and pragmatic relevance.

Our work encompasses several theoretical implications. First, this research extends the scope of the LC4MP model into the context of videoconferencing. Specifically, previous LC4MP studies predominantly addressed individual's one-way information reception, such as viewing advertisements [36, 78, 79]. By expanding LC4MP to videoconferencing research, we examine the two-way interactions among individuals, investigating how people manage constant information exchange and its impact on them. Previous studies on LC4MP have primarily concentrated on the reception of information, overlooking the continuous aspect of responding to information in two-way interactions. While earlier research has indeed examined individuals' responses to information, these responses are often not continuous. Individuals typically receive specific information in a one-way manner, with an emphasis on its persuasiveness rather than the interaction between people [36]. Consequently, this study aims to address the gap in LC4MP by focusing on the real-time interaction between individuals.

Additionally, we found that apart from information overload, communication overload also contributes significant drain on individuals' cognitive resources. In the context of videoconferencing, we emphasize how people's communication needs during videoconferencing interactions influence VF. The results show a positive relationship between communication overload and VF, underscoring the crucial role of verbal communication in videoconferencing. As mentioned earlier, previous research has predominantly focused on the reception of information and explored its effects on individuals rather than delving into synchronous

communication [36, 79]. In videoconferencing, individuals not only need to respond to information immediately, but this response takes place through a continual process during the videoconference session. Throughout this process, individuals undergo the three information processing stages mentioned in LC4MP: encoding, storage, and retrieval. Thus, the longer a user spends videoconferencing, the more frequently they will engage in information processing, potentially leading to the depletion of cognitive resources and resulting in fatigue. Therefore, in the videoconferencing environment, in addition to information overload being a significant source of resource consumption, communication overload also plays a crucial role. Hence, this study introduces another factor in LC4MP that affects cognitive resource consumption.

Finally, while past studies on LC4MP were done in the context of persuasion (e.g. persuasion in video game advertising) [80], social influence (e.g. cancer patient communication, persuasion in video game advertising) [32], and education media [81] and focused on the outcomes of information overload and the mechanisms by which information stimulates emotions [32, 80, 81], this study extends the investigation on LC4MP to the context of videoconferencing. The results further indicate that in an era of advanced technology and widespread media usage, communication styles across different media platforms may exert distinct effects on individuals. In comparison to other social media, synchronous communication, closely resembling face-to-face interaction, may demand more cognitive resources. This aspect warrants further verification in future studies.

The findings of this study offer practical insights into the effective utilization of videoconferencing. Whether for work or educational purposes, it is crucial to consider the frequency of videoconferencing and its potential in causing VF, particularly considering the elevated communication overload users face. Hence, researchers and practitioners should be attuned to the stressors videoconference users' face during videoconferencing, in alleviating VF that users experience. One potential solution is to encourage users who are hesitant to speak up to use text dialog boxes within videoconferencing platforms to communicate, allowing them to express their ideas while reducing potential cognitive overload. Additionally, it is advisable to allocate ample time for users to effectively comprehend and process the information presented during videoconferencing. Moreover, the duration of videoconferencing should be carefully considered, enabling managers to flexibly set meeting times while maintaining videoconferencing efficiency [8].

Despite the contributions, our research has several limitations. Primarily, our cultural dichotomy, contrasting Western and Asian influences on VF from the perspective of individualism and collectivism, represents a mere slice of the broader picture. Future studies could explore the impact of different policies and working styles on VF within the same cultural context, fostering a more comprehensive understanding. In addition, the survey method limits our ability to truly ascertain causal relationships. To gain a more concrete understanding of the causal actors affecting VF, future research should consider using experimental designs. Additionally, in this study, we only considered the different user groups and their usage purposes. However, many other potential factors could also influence VF. These factors include the timing of the videoconference (day vs. night; during vs. after office hours), the location where the videoconferencing is conducted (e.g., at home or in the office), and the number of participants in the videoconferencing. Future research should explore the impact of these variables on VF.

Overall, our study highlights that both communication overload and information overload are significant contributors to VF in different contexts (e.g., educational and professional settings). Specifically, the results indicate that higher levels of these overloads are associated with increased fatigue, supporting our hypotheses. These findings are closely aligned with previous

research on VF. Based on these findings, we have proposed several strategies to alleviate VF. Given the increasing importance of videoconferencing in daily life, future research should examine the causes of VF and potential solutions from various perspectives.

## Acknowledgments

The authors thank Yixuan Ong for assisting with an early draft of the manuscript.

## Author Contributions

**Conceptualization:** Heng Zhang.

**Investigation:** Heng Zhang, Christian Montag.

**Methodology:** Benjamin J. Li, Heng Zhang.

**Resources:** Christian Montag.

**Supervision:** Benjamin J. Li.

**Writing – original draft:** Benjamin J. Li, Heng Zhang.

**Writing – review & editing:** Benjamin J. Li, Heng Zhang, Christian Montag.

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
