## [Decision Letter · Decision Letter 0]

15 Jul 2024

PONE-D-24-13534Too much to process? Exploring the relationships between communication and information overload and videoconference fatiguePLOS ONE

Dear Dr. Li,

Thank you for submitting your manuscript to PLOS ONE. After careful consideration, we feel that it has merit but does not fully meet PLOS ONE’s publication criteria as it currently stands. Therefore, we invite you to submit a revised version of the manuscript that addresses the points raised during the review process.

We look forward to receiving your revised manuscript.

Kind regards,

Shairyzah Ahmad Hisham, PhD.

Academic Editor

PLOS ONE

Journal Requirements:

Additional Editor Comments:

Dear corresponding author,

Congratulations for the completion of this timely and high-quality research. Your manuscript has been reviewed by 2 independent reviewers and now requires minor revision. Please make the necessary amendments based on the comments and recommendations from the reviewers and resubmit according to the timeline provided. Please do not hesitate to contact me if more time is required for the amendments.

Thank you.

Reviewers' comments:

Reviewer's Responses to Questions

**Comments to the Author**

1. Is the manuscript technically sound, and do the data support the conclusions?

Reviewer #1: Yes

Reviewer #2: Yes

2. Has the statistical analysis been performed appropriately and rigorously? 

Reviewer #1: Yes

Reviewer #2: Yes

3. Have the authors made all data underlying the findings in their manuscript fully available?

Reviewer #1: Yes

Reviewer #2: Yes

4. Is the manuscript presented in an intelligible fashion and written in standard English?

Reviewer #1: Yes

Reviewer #2: Yes

5. Review Comments to the Author

**Reviewer #1:** The topic, "Too much to process? Exploring the relationships between communication and information overload and videoconference fatigue," is both timely and significant, offering valuable insights into an increasingly relevant aspect of modern communication. The authors should be commended for their well-structured study, which presents information in a clear and organised manner, facilitating easy navigation through its various sections. The use of headings and subheadings is particularly commendable, as it enhances the readability and coherence of the content.

The chosen methods align effectively with the research question and objectives outlined in the study, demonstrating a thoughtful approach to data collection and analysis. However, there are areas where further refinement could enhance the manuscript's clarity and impact. For instance, while both the introduction and literature review sections provide valuable context, there is some overlap and redundancy in the information presented. By streamlining these sections and merging them into a concise background, the authors can focus more sharply on key studies and their gaps, thereby strengthening the manuscript's overall coherence (Lines 42-64) and (Lines 93-96), (Lines 17-34) and (Lines 24-43, 87-92).

Furthermore, while the data and analyses effectively support the study's claims, the complexity of the analyses may pose a challenge for some readers to follow. Providing a clear roadmap of the analysis process in the methodology section would greatly assist readers in navigating through the steps of data analysis. Additionally, simplifying dense tables and provide more narrative explanation where necessary and ensuring clear labelling of figures will enhance their interpretability and accessibility. It is worth noting that Table 5 is not directly discussed in the provided excerpts. However, it is mentioned that Figure 4 can be accessed or downloaded, which may contain relevant information related to the study. Unfortunately, without the specific content of Table 5, it is challenging to provide a detailed assessment of its density and interpretability.

In the discussion section, the authors have an opportunity to expand on the practical implications of their findings and contextualise them within the broader literature. By discussing specific strategies for reducing videoconference fatigue in different contexts (e.g., educational vs. professional), the authors can offer valuable insights for practitioners and researchers alike. This could include a summarised key finding indicating that both communication overload and information overload significantly contribute to videoconference fatigue. Specifically, the results indicated that higher levels of these overloads are associated with increased fatigue, supporting our hypotheses. These findings align closely with prior research on videoconference fatigue, which has consistently highlighted the detrimental effects of excessive communication and information processing during virtual meetings.

In conclusion, I appreciate the opportunity to review this manuscript and commend the authors for their efforts in conducting and presenting this research. Despite the areas identified for improvement, the manuscript has significant potential to contribute to the field. I encourage the authors to carefully consider the feedback provided and to address the suggested revisions to enhance the clarity, focus, and overall impact of the paper.

**Reviewer #2: **Dear authors,

First and foremost, congratulations for completing the research work and for writing an overall good manuscript. I appreciate how the 2 studies were presented separately which make them easier to read. All statistical analyses were very well reported.

I, however, have a few comments/suggestions for your further considerations:

Abstract

- Line 29-34: As the overall results are similar for both Study 1 & Study 2, consider to combine the sentences.

- Line 34: To add the word 'and' between Information overload and VF.

- To add conclusions for both studies.

Introduction

- To add a description on current curriculum delivery in Singapore i.e., percentage of online and face-to-face classes etc. This is an important information as it may partly help justify why i) secondary school students and not university students were chosen to participate in this study ii) the 4 schools were selected

Methodology(for both studies)

- to add description on Participant/Respondent Information Sheet

- to elaborate on platforms used to distribute the survey instrument

- Line 245 (page 11): To remove the word 'teenagers' as there are participants who are 21 years old

- to move all demographic results to the Results section

- Table 1 (page 13): to describe what the term videoconference means to secondary schools students.

- To add the Inclusion and Exclusion criteria of participants for Study 2.

Results

- Table 1 (page 13): To explain the difference between Genfat1 & Genfat2 and between Emofat1 and Genfat3. You could add the explanation in the results or in the Discussion section.

- Table 1 (page 13): To explain the meaning of 'social situation'. You could add the explanation in the results or in the Discussion section.

- To consider providing a table on demographic profiles for Study 2.

Limitations

- To add potential confounders in this study e.g., time of videoconference (day vs night; during vs after office hours), nature or type of work the participants involved in, device they use for the videoconference, place of videoconferencing (own desk vs large meeting room). All these factors could alternatively be included in the discussions.

All the best and thank you.

6. PLOS authors have the option to publish the peer review history of their article (what does this mean?). If published, this will include your full peer review and any attached files.

Reviewer #1: No

Reviewer #2: **Yes: **ZAINOL AKBAR ZAINAL

---

## [Author Response · Author response to Decision Letter 0]

21 Aug 2024

Additional Editor Comments:

Dear corresponding author,

Congratulations for the completion of this timely and high-quality research. Your manuscript has been reviewed by 2 independent reviewers and now requires minor revision. Please make the necessary amendments based on the comments and recommendations from the reviewers and resubmit according to the timeline provided. Please do not hesitate to contact me if more time is required for the amendments.

Thank you.

Reviewers' comments:

Reviewer #1: The topic, "Too much to process? Exploring the relationships between communication and information overload and videoconference fatigue," is both timely and significant, offering valuable insights into an increasingly relevant aspect of modern communication. The authors should be commended for their well-structured study, which presents information in a clear and organised manner, facilitating easy navigation through its various sections. The use of headings and subheadings is particularly commendable, as it enhances the readability and coherence of the content.

Response: Thank you for your encouraging comments.

Comment: The chosen methods align effectively with the research question and objectives outlined in the study, demonstrating a thoughtful approach to data collection and analysis. However, there are areas where further refinement could enhance the manuscript's clarity and impact. For instance, while both the introduction and literature review sections provide valuable context, there is some overlap and redundancy in the information presented. By streamlining these sections and merging them into a concise background, the authors can focus more sharply on key studies and their gaps, thereby strengthening the manuscript's overall coherence (Lines 42-64) and (Lines 93-96), (Lines 17-34) and (Lines 24-43, 87-92).

Response: Thank you for raising this and we agree with your point. We have refined each paragraph to be more concise and eliminated repetition between them. We have included the following:

For (Lines 42-64) and (Lines 93-96):

“The pandemic catalyzed unprecedented shifts in human communication patterns, with a notable surge in videoconferencing (1–3). This technology has become essential for interactions even after the pandemic, especially when in-person meetings are impractical. Therefore, videoconferencing permeates a variety of global settings, from classrooms and workplaces to friends and family gatherings (4–7). As remote and hybrid models grow, platforms like Microsoft Teams and Zoom have experienced exponential growth (8). Videoconferencing for remote study and work is likely to become common practice in the near future (9–11). 

Videoconferencing enabling face-to-face communication through the transmission of sound, images, and data, making it a form of computer-mediated communication (12). According to the media enrichment theory, the degree to which this digital medium emulates face-to-face communication is directly related to its enrichment (13). From this perspective, videoconferencing stands out as a profoundly enriched communication medium, providing users with abundant nonverbal cues and instantaneous responses to enhance the quality of communication (14). However, it also has drawbacks. Recent studies identify videoconference fatigue (VF), where users feel exhausted, disengaged, and anxious after long sessions, impacting mental health and productivity (15, p. 813–17).”

“As we mentioned earlier, prolonged use of videoconferencing can lead to VF.”

For (Lines 17-34) and (Lines 24-43, 87-92):

“Study One focused on the educational context and comprised a survey with 489 students. In Study Two, we expanded our exploration to the professional use of videoconferencing in two populations: Singapore and Germany. A total of 610 responses were collected in Singapore, with the German sample constituting a total of 948 participants. Analyses from both studies, the results consistently demonstrated a positive relationship between videoconference frequency and communication overload. Additionally, users’ perceived communication overload was positively associated with information overload and VF.”

“Using the LC4MP framework, we investigate how videoconferencing length and frequency impact VF and the mediating roles of perceived information and communication overload. Through two distinct studies, we explored the differences between various groups and the differences under different cultural backgrounds.

Comment: Furthermore, while the data and analyses effectively support the study's claims, the complexity of the analyses may pose a challenge for some readers to follow. Providing a clear roadmap of the analysis process in the methodology section would greatly assist readers in navigating through the steps of data analysis. Additionally, simplifying dense tables and provide more narrative explanation where necessary and ensuring clear labelling of figures will enhance their interpretability and accessibility. It is worth noting that Table 5 is not directly discussed in the provided excerpts. However, it is mentioned that Figure 4 can be accessed or downloaded, which may contain relevant information related to the study. Unfortunately, without the specific content of Table 5, it is challenging to provide a detailed assessment of its density and interpretability.

Response: Thank you for your suggestion. We have moved Table 5 to the Results section because we provide a more detailed description of the table there. In the Data Analysis section, we mainly present the methods used, while the Results section focuses more on displaying the data outcomes.

Comment: In the discussion section, the authors have an opportunity to expand on the practical implications of their findings and contextualise them within the broader literature. By discussing specific strategies for reducing videoconference fatigue in different contexts (e.g., educational vs. professional), the authors can offer valuable insights for practitioners and researchers alike. This could include a summarised key finding indicating that both communication overload and information overload significantly contribute to videoconference fatigue. Specifically, the results indicated that higher levels of these overloads are associated with increased fatigue, supporting our hypotheses. These findings align closely with prior research on videoconference fatigue, which has consistently highlighted the detrimental effects of excessive communication and information processing during virtual meetings.

Response: Thank you very much for pointing this out. We have a more detailed discussion in the discussion and limitation sections. We have included the following:

“Despite the contributions, our research has several limitations. Primarily, our cultural dichotomy, contrasting Western and Asian influences on VF from the perspective of individualism and collectivism, represents a mere slice of the broader picture. Future studies could explore the impact of different policies and working styles on VF within the same cultural context, fostering a more comprehensive understanding. In addition, the survey method limits our ability to probe the causal relationships. To gain a more concrete understanding of the causal actors affecting VF, future research should consider using experimental designs. Additionally, in this study, we only considered the different user groups and their usage purposes. However, many other potential factors could also influence VF. These factors include the timing of the videoconference (day vs. night; during vs. after office hours), the location where the videoconferencing is conducted (at home or in the office), and the number of participants in the videoconferencing. Future research should explore the impact of these variables on VF. 

Overall, our study highlights that both communication overload and information overload are significant contributors to VF in different contexts (e.g., educational and professional settings). Specifically, the results indicate that higher levels of these overloads are associated with increased fatigue, supporting our hypotheses. These findings are closely aligned with previous research on VF. Based on these findings, we have proposed several strategies to alleviate VF. Given the increasing importance of videoconferencing in daily life, future research should examine the causes of VF and potential solutions from various perspectives. This would help to better guide the effective use of videoconferencing.”

Comment: In conclusion, I appreciate the opportunity to review this manuscript and commend the authors for their efforts in conducting and presenting this research. Despite the areas identified for improvement, the manuscript has significant potential to contribute to the field. I encourage the authors to carefully consider the feedback provided and to address the suggested revisions to enhance the clarity, focus, and overall impact of the paper.

Response: We thank you for your very helpful feedback and believe they have helped to enhance the paper.

Reviewer #2: Dear authors,

First and foremost, congratulations for completing the research work and for writing an overall good manuscript. I appreciate how the 2 studies were presented separately which make them easier to read. All statistical analyses were very well reported.

Response: Thank you for your encouraging words.

Comment: I, however, have a few comments/suggestions for your further considerations:

Abstract

- Line 29-34: As the overall results are similar for both Study 1 & Study 2, consider to combine the sentences.

- Line 34: To add the word 'and' between Information overload and VF.

- To add conclusions for both studies.

Response: Thank you very much for your suggestions. Based on your advice, we have rewritten the abstract. We have included the following:

“The adoption of videoconferencing has brought significant convenience to people's lives. However, as videoconferencing usage has skyrocketed, it has unveiled a range of side effects, most notably videoconference fatigue (VF). In response, this paper employed the Limited Capacity Model of Motivated Mediated Message Processing (LC4MP) as a theoretical framework to conduct two comprehensive investigations, centering on the impact of verbal communication overload on users’ information overload and VF. We conducted two studies to test our propositions and conceptual model. Study One focused on the educational context and comprised a survey with 489 students. In Study Two, we expanded our exploration to the professional use of videoconferencing in two populations: Singapore and Germany. A total of 610 responses were collected in Singapore, with the German sample constituting a total of 948 participants. Analyses from both studies, the results consistently demonstrated a positive relationship between videoconference frequency and communication overload. Additionally, users’ perceived communication overload was positively associated with information overload and VF. Due to the different purposes of using videoconferencing, Study Two indicates that videoconference fatigue reduces information overload. Based on the findings of the two studies, we discussed the theoretical and practical implications and suggested new directions for videoconferencing research.”

Comment:

Introduction

- To add a description on current curriculum delivery in Singapore i.e., percentage of online and face-to-face classes etc. This is an important information as it may partly help justify why i) secondary school students and not university students were chosen to participate in this study ii) the 4 schools were selected

Response: Thank you for this. We have included the following:

“Moreover, among the student population, we focus specifically on secondary school students. The reasons are as follows: firstly, according to reports from the Singaporean education authorities, approximately 96% of secondary school students participated in full home-based learning during the COVID-19 pandemic (53). Compared to university students, secondary school students had almost no prior experience with online courses before the pandemic. Many university students might have had online education experience before COVID-19, so attending classes via videoconferencing is not unfamiliar to them, but it is a completely new experience for secondary school students. Secondly, secondary school students have more intensive course schedules compared to university students. They need to use videoconferencing regularly every day for their classes, making them more prone to VF. To address technical challenges, the Singaporean government provided each secondary school student with a tablet, whereas university students typically already have their own electronic devices (54). In summary, using videoconferencing as a teaching method is more challenging for secondary school students. Additionally, to ensure a more representative sample, we used the snowball sampling method and selected four different schools in Singapore.”

Comment: 

Methodology (for both studies)

- to add description on Participant/Respondent Information Sheet

Response: Thank you for this. We have included the following:

“Before the survey began, all participants were given sufficient time to read the Information Sheet and sign it. During the experiment, all participants had the right to withdraw at any time or choose not to answer any questions that made them feel uncomfortable.”

Comment: - to elaborate on platforms used to distribute the survey instrument

Response: Thank you very much for pointing this out. We have included the following:

“Qualtrics is a highly regarded and extensively utilized survey platform that supports the comprehensive creation, distribution, and analysis of surveys (53). It offers an online format, making it easily accessible to participants using various digital devices, including computers, tablets, and smartphones.”

Comment: - Line 245 (page 11): To remove the word 'teenagers' as there are participants who are 21 years old

Response: Thank you very much for pointing this out. We removed the word 'teenagers' and changed it to “students”. The sentence now reads:

“An online nationwide survey via Qualtrics was conducted from 3 October to 30 November 2022 with 489 students from four secondary schools in Singapore.”

Comment: - to move all demographic results to the Results section

Response: Thank you for raising this. We moved all demographic results to the Results section (both study 1 and 2) . Please check the latest version of the manuscript.

Comment: - Table 1 (page 13): to describe what the term videoconference means to secondary schools students.

Response: Thanks to your suggestion, we have revised the paper based on your suggestion. The following has been included:

“Before the questionnaire began, we described videoconferences as instead of meeting face-to-face in a classroom or a physical location, you can see and hear each other through their computer screens, tablets, or smartphones. Like Zoom, Teams, etc. that you're using during covid-19”

Comment: - To add the Inclusion and Exclusion criteria of participants for Study 2.

Response: Thank you very much for pointing this out. 

“All participants were over 21 years old and had prior experience using videoconferencing. Participants who did not meet these criteria were not included in the study population for both countries.”

Comment: 

Results

- Table 1 (page 13): To explain the difference between Genfat1 & Genfat2 and between Emofat1 and Genfat3. You could add the explanation in the results or in the Discussion section.

- Table 1 (page 13): To explain the meaning of 'social situation'. You could add the explanation in the results or in the Discussion section.

Response: Thank you for this comment. We understand that the items may need clarif

---

## [Decision Letter · Decision Letter 1]

20 Sep 2024

PONE-D-24-13534R1Too much to process? Exploring the relationships between communication and information overload and videoconference fatiguePLOS ONE

Dear Dr. ZHANG,

Thank you for submitting your revised manuscript to PLOS ONE. The manuscript has addressed all the points raised by the reviewers. However, to further improve the quality of the manuscript, please conduct proofreading as recommended by one of the reviewers below.

We look forward to receiving your revised manuscript.

Kind regards,

Shairyzah Ahmad Hisham, PhD.

Academic Editor

PLOS ONE

Journal Requirements:

Reviewers' comments:

Reviewer's Responses to Questions

**Comments to the Author**

1. If the authors have adequately addressed your comments raised in a previous round of review and you feel that this manuscript is now acceptable for publication, you may indicate that here to bypass the “Comments to the Author” section, enter your conflict of interest statement in the “Confidential to Editor” section, and submit your "Accept" recommendation.

Reviewer #1: All comments have been addressed

Reviewer #2: All comments have been addressed

2. Is the manuscript technically sound, and do the data support the conclusions?

Reviewer #1: Yes

Reviewer #2: Yes

3. Has the statistical analysis been performed appropriately and rigorously? 

Reviewer #1: Yes

Reviewer #2: Yes

4. Have the authors made all data underlying the findings in their manuscript fully available?

Reviewer #1: Yes

Reviewer #2: Yes

5. Is the manuscript presented in an intelligible fashion and written in standard English?

Reviewer #1: Yes

Reviewer #2: No

6. Review Comments to the Author

Reviewer #1: Dear Authors,

Thank you for the opportunity to review your manuscript titled "Too much to process? Exploring the relationships between communication and information overload and videoconference fatigue." Your study addresses a highly relevant and timely issue in the context of modern communication, particularly as the world increasingly relies on videoconferencing for both educational and professional interactions. Overall, the manuscript is well-structured, methodologically sound, and written in clear and standard English.

Reviewer #2: Dear authors,

Thank you for considering my previous suggestions and for addressing all comments I made in the original manuscript. All comments have been addressed accordingly and are acceptable.

I, however, would strongly recommend that the revised manuscript to go through another round of proofreading to enhance the clarity of the sentences, especially for the newly added sentences.

Please kindly refer to some examples below:

- Line 266 (page 11): 'who under 21 old'

- Line 268 (page 12): 'all participants had the right to withdraw'  to consider rephrasing the sentence to 'all participants were made aware about their rights to withdraw'

- Line 309 (page 15): 'As Singapore is a multiracial society'  to consider using the word 'country' instead of 'society'.

Thank you and all the best.

7. PLOS authors have the option to publish the peer review history of their article (what does this mean?). If published, this will include your full peer review and any attached files.

Reviewer #1: No

Reviewer #2: **Yes: **ZAINOL AKBAR ZAINAL

---

## [Author Response · Author response to Decision Letter 1]

24 Sep 2024

Comments to the Author

Reviewer #1: Dear Authors,

Thank you for the opportunity to review your manuscript titled "Too much to process? Exploring the relationships between communication and information overload and videoconference fatigue." Your study addresses a highly relevant and timely issue in the context of modern communication, particularly as the world increasingly relies on videoconferencing for both educational and professional interactions. Overall, the manuscript is well-structured, methodologically sound, and written in clear and standard English.

Response: Thank you for your positive comments.

Reviewer #2: Dear authors,

Thank you for considering my previous suggestions and for addressing all comments I made in the original manuscript. All comments have been addressed accordingly and are acceptable.

I, however, would strongly recommend that the revised manuscript to go through another round of proofreading to enhance the clarity of the sentences, especially for the newly added sentences.

Please kindly refer to some examples below:

- Line 266 (page 11): 'who under 21 old'

- Line 268 (page 12): 'all participants had the right to withdraw'  to consider rephrasing the sentence to 'all participants were made aware about their rights to withdraw'

- Line 309 (page 15): 'As Singapore is a multiracial society'  to consider using the word 'country' instead of 'society'.

Response: Thank you very much for your suggestions. We have proofread the article closely and made edits to improve the clarity of the manuscript. In addition, we have made the following edits based on the examples you raised:

“Participation was entirely voluntary. All respondents provided their informed consent prior to data collection, and consent was also obtained from the parents of respondents who were under the age of 21.”

“During the experiment, all participants were made aware about their rights to withdraw at any time or choose not to answer any questions that made them feel uncomfortable.

“As Singapore is a multiracial country”

---

## [Editor Report · Decision Letter 2]

7 Oct 2024

Too much to process? Exploring the relationships between communication and information overload and videoconference fatigue

PONE-D-24-13534R2

Dear Dr. ZHANG,

We’re pleased to inform you that your manuscript has been judged scientifically suitable for publication and will be formally accepted for publication once it meets all outstanding technical requirements.

Kind regards,

Shairyzah Ahmad Hisham, PhD.

Academic Editor

PLOS ONE

---

## [Editor Report · Acceptance letter]

6 Nov 2024

PONE-D-24-13534R2 

PLOS ONE

Dear Dr. ZHANG, 

I'm pleased to inform you that your manuscript has been deemed suitable for publication in PLOS ONE. Congratulations! Your manuscript is now being handed over to our production team.

Kind regards, 

on behalf of

Dr. Shairyzah Ahmad Hisham 

Academic Editor

PLOS ONE